# Antibiotic Use before, during, and after Seeking Care for Acute Febrile Illness at a Hospital Outpatient Department: A Cross-Sectional Study from Rural India

**DOI:** 10.3390/antibiotics11050574

**Published:** 2022-04-25

**Authors:** Bronwen Holloway, Harshitha Chandrasekar, Manju Purohit, Ashish Sharma, Aditya Mathur, Ashish KC, Leticia Fernandez-Carballo, Sabine Dittrich, Helena Hildenwall, Anna Bergström

**Affiliations:** 1Department of Women’s and Children’s Health, Uppsala University, 751 85 Uppsala, Sweden; harshitha.chandrasekar.1957@student.uu.se (H.C.); ashish.k.c@kbh.uu.se (A.K.); anna.bergstrom@kbh.uu.se (A.B.); 2Department of Pathology, Ruxmaniben Deepchand Gardi Medical College, Ujjain 456006, India; manjuraj.purohit64@gmail.com; 3Department of Global Public Health, Health Systems and Policy, Karolinska Institute, 171 77 Stockholm, Sweden; helena.hildenwall@ki.se; 4Department of Medicine, Ruxmaniben Deepchand Gardi Medical College, Ujjain 456006, India; ashishricha2001@yahoo.co.in; 5Department of Paediatrics, Ruxmaniben Deepchand Gardi Medical College, Ujjain 456006, India; dr.adityamathur121@gmail.com; 6Foundation for Innovative New Diagnostics, 1202 Geneva, Switzerland; leticia.fernandez@finddx.org (L.F.-C.); sabine.dittrich@finddx.org (S.D.); 7Astrid Lindgren Children’s Hospital, Karolinska University Hospital, 171 64 Solna, Sweden; 8Department of Clinical Science, Intervention and Technology, Karolinska Institute, 141 52 Huddinge, Sweden

**Keywords:** antibiotics, antibiotic use, antibiotic resistance, acute febrile illness, fever, AWaRe, access, excess, rural population, India, low- and middle-income countries

## Abstract

Antibiotic resistance is a naturally occurring phenomenon, but the misuse and overuse of antibiotics is accelerating the process. This study aimed to quantify and compare antibiotic use before, during, and after seeking outpatient care for acute febrile illness in Ujjain, India. Data were collected through interviews with patients/patient attendants. The prevalence and choice of antibiotics is described by the WHO AWaRe categories and Anatomical Therapeutic Chemical classes, comparing between age groups. Units of measurement include courses, encounters, and Defined Daily Doses (DDDs). The antibiotic prescription during the outpatient visit was also described in relation to the patients’ presumptive diagnosis. Of 1000 included patients, 31.1% (*n* = 311) received one antibiotic course, 8.1% (*n* = 81) two, 1.3% (*n* = 13) three, 0.4% (*n* = 4) four, 0.1% (*n* = 1) five, and the remaining 59.0% (*n* = 590) received no antibiotics. The leading contributors to the total antibiotic volume in the DDDs were macrolides (30.3%), combinations of penicillins, including β-lactamase inhibitors (18.8%), tetracyclines (14.8%), fluoroquinolones (14.6%), and third-generation cephalosporins (13.7%). ‘Watch’ antibiotics accounted for 72.3%, 52.7%, and 64.0% of encounters before, during, and after the outpatient visit, respectively. Acute viral illness accounted for almost half of the total DDDs at the outpatient visit (642.1/1425.3, 45.1%), for which the macrolide antibiotic azithromycin was the most frequently prescribed antibiotic (261.3/642.1, 40.7%).

## 1. Introduction

Antibiotic resistance is a naturally occurring phenomenon, but the misuse of antibiotics is accelerating the process. India, the world’s second-most populous country, is the world’s largest consumer of antibiotics based on total volume [1]. The level of antibiotic resistance across India is rising [1], and while there are a lack of comprehensive nationwide representative data, the recent antimicrobial susceptibility profile of the Global Antimicrobial Resistance Surveillance System (GLASS) highlights high resistance in both Gram-negative and Gram-positive organisms [2]. To limit the development of antibiotic resistance, the right drugs must be used for the right reasons, and treatment options should be targeted at those least likely to build resistance. To support the monitoring and appropriate use of antibiotics, the World Health Organization (WHO) introduced the AWaRe classification system: the ‘Access’, ‘Watch’, ‘Reserve’ and ‘Not Recommended’ groups are based on their use as first-, second- or last-choice treatment options, their spectrum of activity, and their susceptibility to induce resistance in organisms [3].

Increasingly, health services are delivered through mixed systems of public and private providers [4]. In India, approximately 70% of outpatient care is provided by the private sector [4], mainly concentrated in urban areas [5]. In rural areas, healthcare provision has shifted to drug retailers and informal healthcare providers [6,7] and patients often seek care and treatment from several outlets during a single episode of illness [8,9]. Despite government regulations which require a valid prescription, most antibiotics are readily available and the illegal over-the-counter purchase of antibiotics is widespread [10,11]. Antibiotic use across the diversity of healthcare providers and informal markets in low- and middle-income countries (LMICs) is not systematically monitored. A handful of studies from LMICs have examined antibiotic use prior to seeking care for fever, together with an antibiotic prescription as part of the patient visit, but none of these studies explored antibiotic use after care from outside the primary care/hospital setting [12,13,14,15].

Understanding how antibiotics are used across the course of illness is necessary to identify areas in need of improvement and to develop context-appropriate interventions to optimize antibiotic use. To assess and compare antibiotic use throughout the course of acute febrile illness (AFI), a secondary analysis was performed on data collected from an observational study on the cause and management of acute febrile illness in children and adult outpatients in India. The aim of this analysis was to quantify and compare antibiotic use prior to study inclusion, antibiotic prescription during the outpatient visit, and antibiotic use during a seven-day follow-up period, by AWaRe classification [16] and Anatomical Therapeutic Chemical class [17].

## 2. Materials and Methods

### 2.1. Study Design and Setting

A secondary analysis of data generated by a cross-sectional study investigating the cause and management of AFI (Clinical Trials Registry—India; REF/2019/05/026081). All patients from the original study were included in the analysis. The study took place at the pediatric and medicine outpatient departments of R.D. Gardi Medical College, a tertiary-teaching hospital in rural Ujjain, India. The climate is tropical, with a rainy season from July to September, characterized by the simultaneous occurrence of many fever-causing diseases (i.e., malaria, typhoid, respiratory tract infection, diarrhea, dengue, etc.). Antibiotics can be obtained at pharmacies and drug shops or may be provided as part of the care received from a visit to an informal healthcare provider [18].

### 2.2. Patient Population

Inclusion criteria were children (2 months–17 years) and adult (18–65 years) outpatients with AFI, defined as an axillary temperature ≥ 37.5 °C, measured electronically at initial evaluation or a history of fever in the previous 48 h, a duration ≤ 14 days, and no signs of severe illness.

### 2.3. Data Collection Procedure

Patients were recruited from June to August 2019. All patients presenting to the pediatric and medicine outpatient departments were clinically managed as per routine care by a resident physician in training or specialized in pediatrics or general medicine. For the purposes of this study, the resident physician made a presumptive diagnosis. This diagnosis was a free text description based on clinical assessment during the outpatient visit, without the support of any diagnostic tests, following standard clinical practice. After consultation, patients were screened for fever, assessed for inclusion, and enrolled in the study. Patients were requested to return for follow-up one week after enrolment; those not reporting by day seven were called by phone until they could be reached. Study activities did not interfere with the treatment of AFI or antibiotic prescription as part of routine practice.

Data were collected on antibiotic use throughout the course of AFI: before, during, and after visit to the hospital outpatient department. “Before” was defined as a patient/patient attendant reporting antibiotic use for the current episode of illness (within the last 14 days) prior to presenting to the outpatient department. “During” included antibiotic prescriptions received from the resident physician as part of routine practice at the outpatient visit prior to study inclusion. “After” was defined as additional patient-/patient attendant-reported antibiotic use within the seven-day follow-up period.

Using standardized case report forms, a dedicated study clinician collected the presumptive diagnosis and antibiotic prescription by the resident physician from the patient’s outpatient record. Following this, the study clinician interviewed the patients/patient attendants regarding reported antibiotic use before/after the outpatient visit. Patients/patient attendants were asked to name the antibiotics, show the medicines or their packages or send a picture via text message. They were also asked to describe the route, duration, dose, and frequency of their treatment. Antibiotics before/after the outpatient visit could have been obtained over the counter from a pharmacy or drug shop, or from an informal healthcare provider, or another formal healthcare provider outside the study hospital. Antibiotic use after the outpatient visit could have also been provided by a resident physician at the study hospital during the follow-up visit. Patients/patient attendants who reported antibiotic use at follow-up were asked if it was done by a formal/informal provider, or self-medication. Data were also collected on fever duration in days at enrolment, and the time to resolution of illness in days, as reported by patients at follow-up.

### 2.4. Data Analysis

Any antibiotics in the Anatomical Therapeutic Chemical (ATC) classes J01-antibacterial agents for systemic use, A07-intestinal antibiotics, and orally administered metronidazole (P01AB01), were considered [17]. Antibiotic use was described before, during, and after an outpatient visit by age group in years (children < 5, 5–17 years; adults 18–34, 35–49, 50–65 years). Prescriptions during the outpatient visit were also described in relation to the presumptive diagnosis provided by the resident physician. The frequency and choice of antibiotic were described by number of antibiotic courses, AWaRe classification [16] in antibiotic encounters, and ATC class (as defined by the ATC 4th level) in Defined Daily Doses (DDDs) [19]. The term antibiotic course is used to account for each unique type of antibiotic (or unique combination of antibiotics) irrespective of the dose, frequency, or duration. An encounter is defined as a patient receiving one or more antibiotics at any of the three-time points (i.e., a patient can have a maximum of three encounters). For encounters with more than one antibiotic, the AWaRe classification was based on a higher restricted antibiotic, i.e., an encounter with both a ‘Watch’ and ‘Reserve’ antibiotic would be classified as ‘Reserve’ [20]. The DDDs were calculated using dosage, frequency, duration of the prescribed antibiotic(s), and the DDD assigned for each drug by the WHO Collaborating Center and the WHO International Working Group on Drug Statistics Methodology [19]. DDDs were applied for both adults and children, as specified DDDs for children have not been created yet (Appendix A) [17,20]. Irrational fixed-dose combinations (FDCs) refer to two or more actives in a fixed proportion of doses with a lack of proven efficacy that are not approved by regulatory agencies [21,22,23]. The irrational FDC of cefixime and azithromycin has no ATC code or DDD; for the purposes of this study, it was treated as other antibiotics in the “Combinations of antibacterials” subgroup. The irrational FDC of norfloxacin and tinidazole was grouped as ‘Unclassified’ as it is not on the AWaRe list. Data were analyzed using STATA (version 16; Stata Corp., College Station, TX, USA). The Chi-square test or Fisher’s exact test was used to compare proportions; a significant level of *p* < 0.05 was used.

## 3. Results

A total of 7551 patients presented to the pediatric and medicine outpatient departments throughout the study period (June to August 2019), of which 504 children and 496 adults were enrolled in the study, all patients were reached at follow-up. Of all 1000 patients, 31.1% (*n* = 311) received one antibiotic course, 8.1% (*n* = 81) two, 1.3% (*n* = 13) three, 0.4% (*n* = 4) four, 0.1% (*n* = 1) five, and the remaining 59.0% (*n* = 590) received no antibiotics throughout the episode of illness. In summary, a total of 533 antibiotic courses were reported across 410 (41.0%) of the 1000 patients. A total of 24 different antibiotics were reported. The leading contributors of the total antibiotic volume in the DDDs were macrolides (30.3%), combinations of penicillins, including β-lactamase inhibitors (18.8%), tetracyclines (14.8%), fluoroquinolones (14.6%), and third-generation cephalosporins (13.7%). The patient demographic and clinical characteristics are presented in Table 1.

### 3.1. Patient-Reported Antibiotic Use before Seeking Care at the Outpatient Department

Overall, 8.3% (83/1000) of patients reported the use of antibiotics before their outpatient visit. Among them, 91.6% (76/83), 7.2% (6/83), and 1.2% (1/83) used one, two and three antibiotics, respectively, for a total of 91 antibiotic courses. Of all courses, 6.6% (6/91) were parenteral formulations. Of all 1000 fever patients, fewer children received antibiotics than adults (25/504 [5.0%] vs. 58/496 [11.7%], *p* < 0.001).

Almost three-quarters of antibiotic encounters before the outpatient visit belonged to the ‘Watch’ group (60/83, 72.3%) of antibiotics, with just one quarter (21/83, 25.3%) belonging to the ‘Access’ group, and the remaining (2/83, 2.4%) to the ‘Not Recommended’ category. ‘Watch’ antibiotics were most frequent in ages 18–34 years (29/36, 80.6%) (Figure 1, Appendix A).

The reported antibiotic use before the outpatient visit amounted to 234.2 DDDs. Fluoroquinolones accounted for the highest volume of DDDs (83.6/234.2, 35.7%), dominated by ciprofloxacin and ofloxacin. This was followed by third-generation cephalosporins (54.7/234.2, 23.1%), with cefixime and cefpodoxime as the leading agents in this class. The remainder of the top 90% of DDDs was composed of macrolides [azithromycin] (35.0/234.2, 15.0%), tetracyclines [doxycycline] (28.0/234.2, 12.0%), and combinations of penicillins, including β-lactamase inhibitors [amoxycillin and clavulanic acid] (14.7/234.2, 6.3%). Fluoroquinolones accounted for 17.7% (7.5/42.4) of DDDs among children of all ages. Irrational FDCs were only seen in young children, but accounted for 22.2% (5.0/22.5) of DDDs in ages < 5 years (Figure 2, Appendix A).

### 3.2. Antibiotic Prescription during the Outpatient Visit

Of all 1000 fever patients, 313 patients (31.3%) were prescribed an antibiotic at the outpatient department, 11.2% (35/313) of which had reported already receiving an antibiotic before seeking care. Of these patients, 31.4% (11/35) were prescribed the same antibiotics they had reported already taking, and the remaining 68.6% (24/35) had a different antibiotic prescribed. Fewer children received antibiotics than adults (102/504 [20.2%] vs. 211/496 [42.5%], *p* < 0.001). Of the prescriptions, 90.4% (283/313) contained one antibiotic and 9.6% (30/313) two antibiotics. Half (15/30) of the prescriptions with two antibiotics were for both doxycycline and metronidazole, and 23.3% (6/30) were for ciprofloxacin and metronidazole. Of the total 343 antibiotics courses recorded, 7.6% (26/343) were parenteral formulations.

Of all antibiotic encounters at the outpatient visit, there were slightly more ‘Watch’ than ‘Access’ antibiotics (165/313 [52.7%] vs. 142/313 [45.4%]). There was one encounter of ‘Reserve’ antibiotics, four ‘Not Recommended’, and one ‘Unclassified’. The proportion of ‘Watch’ antibiotics was highest in ages < 5 and decreased gradually with ages up to 50–65 years (15/28, 53.6% to 9/38, 23.7%) (Figure 1, Appendix A).

Antibiotic prescriptions by the resident physician resulted in 1425.3 DDDs. Macrolides [azithromycin] accounted for the greatest proportion (391.1/1425.3, 27.7%), followed by combinations of penicillins, including β-lactamase inhibitors [amoxycillin and clavulanic acid] (314.9/1425.3, 22.3%), tetracyclines [doxycycline] (276.0/1425.3, 19.5%), third-generation cephalosporins (183.3/1425.3, 12.6%), and fluoroquinolones (150.2/1425.3, 10.6%), together which made up over 90% of the DDDs. The proportion of third-generation cephalosporins was greater among children than adults (76.4/309.4 [24.7%] vs. 106.9/1115.9 [9.6%], *p* < 0.0001). Fluoroquinolone use made up 5.0% (15.5/309.4) of the DDDs in children of all ages (Figure 2, Appendix A).

### 3.3. Patient-Reported Antibiotic Use after the Outpatient Visit

Antibiotic use within the seven-day follow-up period was reported by 8.9% (89/1000) of patients. Of them, 88.8% (79/89) reported using one antibiotic and 11.2% (10/89) two antibiotics. Table 2 shows the combined reported antibiotic use and antibiotic prescription before, during, and after the hospital visit, throughout the course of illness. The majority of patients who reported antibiotic use at follow-up sought care from an informal healthcare provider 73.0% (65/89), while 21.3% (19/89) had their treatment modified by a formal provider, and 5.6% (5/89) modified the treatment themselves. Adults were more likely to receive follow-up care from an informal provider (41/51, [80.4%] vs. 24/38, [63.2%], *p* = 0.005) than children. Among all antibiotic courses reported at follow-up, 11.1% (11/99) were parenteral formulations.

Approximately two-thirds (57/89, 64.0%) of all antibiotic encounters at follow-up belonged to ‘Watch’ and one-third (29/89, 32.6%) to ‘Access’. The remaining (3/89, 3.4%) were ‘Unclassified’ and all three were from informal healthcare providers. There was no significant difference in the proportion of ‘Access’ to ‘Watch’ encounters between patients who received care from informal healthcare providers, formal healthcare, or self-medication (20/62 [32.3%], 7/19 [36.8%], 2/5 [40.0%], respectively, *p* = 0.891) (Figure 1, Appendix A).

Over half of the total 394.5 DDDs at follow-up were contributed by macrolides [azithromycin] (195.7/394.5, 52.5%), followed by combinations of penicillins, including β-lactamase inhibitors [amoxycillin and clavulanic acid] (56.5/394.5, 15.3%), fluoroquinolones (66.4/394.5, 12.3%), and third-generation cephalosporins (43.2/394.5, 11.7%), which together comprised over 90% of the DDDs at follow-up. Fluoroquinolones accounted for 15.6% (17.2/110.4) of the DDDs among children of all ages (Figure 2, Appendix A).

### 3.4. Antibiotic Prescription by Presumptive Diagnosis at Outpatient Visit

Of all patients with the seven most common presumptive diagnoses, acute viral illness had the lowest proportion of antibiotics prescribed (132/601, 22.0%), followed by gastroenteritis (22/58, 37.9%), then typhoid (16/40, 40.0%), malaria (7/15, 47.6%), and a urinary tract infection (63/117, 53.8%). Of patients who received antibiotics, the diagnoses with the highest proportion of ‘Watch’ were typhoid (10/24, 62.5%) and acute viral illness (79/132, 59.8%) (Figure 3, Appendix A).

Acute viral illness accounted for almost half of the total DDDs (642.1/1425.3, 45.0%), with azithromycin contributing the largest proportion (261.3/642.1, 40.7%). For urinary tract infections (303.1/1425.3, 21.3%), the greatest proportion of DDDs came from doxycycline (92.0/303.1, 30.4%), all of which was for adult females, and all but one case was given in combination with metronidazole. Both upper respiratory tract infections (URTIs) (163.7/1425.3, 11.5%) and lower respiratory tract infections (LRTIs) (86.5/1425.3, 6.1%), were treated with amoxycillin and clavulanic acid (63.7/163.7 [38.9%] URTI, 34.8/86.5 [40.1%] LRTI), and azithromycin (54.5/163.7 [33.3%] URTI, 39.2/86.5 [45.2%] LRTI). For gastroenteritis (79.9/1425.3, 5.6%), fluoroquinolones were the most frequently prescribed (24.0/79.9, 30.0%). Typhoid (89.7/1425.3, 6.3%) was treated mostly by doxycycline (24.0/89.7, 26.8%) (Figure 4, Appendix A).

## 4. Discussion

The analysis of antibiotic use showed that 41% of patients had an antibiotic at some point throughout the episode of AFI in rural India. A total of 8%, 31%, and 9% of patients received antibiotics before, during, and after seeking care for AFI at a hospital outpatient department. Examining antibiotic use before, during, and after the outpatient visit gives a more complete picture of antibiotic use throughout the course of AFI. This study identified an additional 10% of patients reporting antibiotic use before or after the outpatient visit that would not have been seen by pure outpatient prescribing estimates. The combined volume of DDDs used before (234.2) and after (394.5) the visit accounted for almost one-third of the total DDD volume (628.7/2054.0, 31%) of the study.

Antibiotic use before seeking outpatient care for AFI was similar to reports of 5% from Thailand and Myanmar [15], and 8% in Tanzania [14], but less than the 42% reported in Bangladesh and Nepal [24], and 46% in Pakistan [24]. The variation between settings could be due to differences in epidemiology, access to antibiotics, and severity and length of the fever, compared to the other two studies. At the outpatient visit, the 20% prescription rate for children was lower than observed in a study from China (32%) [25]. A prescription for all ages of 31% was about half of the 61% observed in patients with a suspected infectious etiology in the same study setting, ten years prior [26]. Co-authors from the study setting suggest the substantial reduction in antibiotic prescribing may be a result of a long period of interest and dedication to antimicrobial stewardship at the study hospital.

In 2019, the WHO adopted a country-level target indicator that at least 60% of antibiotic consumption should come from the ‘Access’ group [3]. The 25–45% of ‘Access’ antibiotics identified across the course of illness corresponds to an analysis of oral antibiotic wholesale data, which found India among the countries with the lowest proportion of use of ‘Access’ (35%), highest use of ‘Watch’ (47%), and considerable use of ‘Unclassified’ antibiotics (17%) [27]. The widespread use of ‘Watch’ antibiotics by all types of providers within the follow-up period may be a result of providers choosing second-line treatments for patients who experienced a continuation or worsening of their symptoms. This may also be reflected in a study of antibiotic prescription for severely ill inpatients from the study hospital which reported an average 61% of ‘Access’, with the proportion of ‘Watch’ increasing over the period of 2008–2017 [28]. Affordable and accurate diagnostics which couple pathogen identification to resistance profiles may help providers to make treatment decisions and increase the proportion of ‘Access’ antibiotics [29,30].

The use of a limited number of ‘Reserve’, ‘Not Recommended’ or ‘Unclassified’ antibiotics were identified. Most ‘Reserve’ antibiotics are parental formulations which may be more common with inpatients or patients with severe illness. All four ‘Not Recommended’ or ‘Unclassified’ antibiotics were irrational FDCs which should be avoided due to their lack of proven efficacy and potential effect on selecting for resistance [23]. The 2–3% at each time period was less than the 12% reported in India in a study by Sulis et al. [31]. The lower prevalence found in our study may be due to India’s ban prohibiting the manufacture, sale and distribution of 344 irrational FDCs in 2018 [32]. The lack of ATC codes, DDDs, and AWaRe classifications for some irrational FDCs poses a challenge to drug utilization monitoring and research. It is essential that all drug use can be appropriately quantified and included using standardized methods [17].

Almost half of the antibiotic prescriptions were for patients with a presumed diagnosis of acute viral illness. This might reflect uncertainty of diagnosis in the absence of diagnostic tests and “just in case” prescribing. A surprising proportion of patients with urinary tract infections and LRTI had no antibiotics prescribed. For UTIs, physicians may have refrained from prescribing until receiving culture results. Nonetheless, the observation of both over and under prescribing reflects the access vs. excess dilemma. While in many cases there is overuse of antibiotics, some patients simultaneously suffer from a lack of antibiotics when needed. The unnecessary use of antibiotics is accompanied with the risk for adverse events, out of pocket expenses, and can cause dangerous diagnostic and treatment delays. Both the quantity and choice of antibiotics is concerning as several alternative antibiotic agents were chosen over the Indian national standard treatment guidelines (STGs) [33]. For urinary tract infections, none of the antibiotics which were major contributors to the total DDDs are suggested as first- or second-line options according to the STGs [33]. Both URTIs and LRTIs were treated predominately with first-line treatment amoxycillin and clavulanic acid, but some cases were treated with azithromycin, which should be reserved as the second-line treatment for LRTIs [33]. In cases of acute gastroenteritis, the STGs advise against antibiotics, and even in cases of presumed bacterial causes, neither the fluoroquinolones nor the β-lactamase inhibiting penicillin chosen are recommended as first-line treatments [33]. For typhoid, empirical treatment is advised for patients with suggestive symptoms and fever for three to four days in endemic areas [34]. Only a portion of the patients with suspected typhoid were prescribed antibiotics, half of which were not recommended by the STGs [33]. The use of doxycycline or ciprofloxacin plus metronidazole may have been an attempt to cover a few of the suspected causes of diarrhea which can require antibiotic therapy such as *V. Cholerae*, *Shigella*, amoebiasis, and giardiasis, as recommended by the STGs [33]. However, these drugs were used across the different diagnoses, not just for gastroenteritis, signaling their likely inappropriate use.

Azithromycin attributed approximately one-third of the total volume of DDDs across all three time periods (621.6/2054.0, 30%). Its low cost and once daily oral formulation makes it one of the most frequently prescribed antibiotics for outpatients in India [31]. The extensive use of azithromycin has been demonstrated to select for macrolide-resistant strains of *Streptococcus pneumoniae* [35], and an increase of macrolide-resistant gene expression in gut microbiota [36]. The extensive use of macrolides when not indicated may unnecessarily select for resistance, rendering important antibiotics no longer as viable treatment options.

Fluoroquinolones are an important class of antibiotics traditionally effective for urinary, respiratory, intra-abdominal, and sexually transmitted infections. As a result of their widespread global use in human health and agriculture, resistance is rising and fluoroquinolones have been detected in surface water and groundwater, posing a serious threat as they are known to be toxic to plants and aquatic organisms [37,38]. The fluoroquinolones use in children under five, before and after an outpatient visit, is alarming considering that fluoroquinolones should be avoided in children due to risks to joint and bone development, and suggests that important training gaps may exist outside of the hospital setting.

### Strengths and Limitations

To our knowledge, this is the first study from a LMIC to examine outpatient antibiotic use throughout the course of AFI which could account for the wide range of sources where antibiotics are available in these settings. The monitoring of drug utilization by an ATC class helps to scrutinize the appropriateness of prescribing at a granular level in relation to treatment guidelines, while the WHO AWaRe classification system makes complex prescribing data easier to understand in relation to antibiotic resistance and, therefore, may be a useful tool for policymakers. R.D. Gardi Medical College Hospital has a long tradition of research on antibiotic use, providing a unique opportunity to compare prescribing patterns between departments, patient groups, and over time.

There are several limitations in this study: First, antibiotic use before and after outpatient visits were based on patient reports and may, thus, be subject to bias. At the study hospital, patients often brought with them any medicines they were currently using. At follow-up, patients contacted by phone were asked to send pictures of their medicines via text message. This likely helped to ensure the correct drug utilization data were collected and increase the robustness of the data. However, it is not possible to know if any antibiotics were overlooked, if the patients adhered to antibiotic use as per the recommended schedule, or if their prescribed treatment was discontinued or switched to another antibiotic by follow-up. Second, the short period of data collection likely missed the seasonality of diseases and the related prescribing practices which would have been available had a full year had been covered. Third, an oversight of the study was the failure to ask the source of antibiotics before the outpatient visit, and including such data in future drug utilization studies in LMICs is recommended. Fourth, it would have been interesting to see if patients with a presumptive diagnosis of malaria were also prescribed antimalarials; unfortunately, however, we did not collect this data. Fifth, using adult DDDs in children may underestimate the actual antibiotic use children, therefore, we presented antibiotic use data by courses, encounters and DDDs to provide different estimates of use. Finally, differences in seasonality, types of infections, antibiotic policies and resistance patterns may make it difficult for a direct comparison of the study results with other settings.

## 5. Conclusions

The overuse and misuse of antibiotics within and outside the hospital outpatient department are of major concern. Interventions to support providers in aligning prescribing practices with the WHO AWaRe recommendations and national standard treatment guidelines are urgently needed.

## Figures and Tables

**Figure 1 antibiotics-11-00574-f001:**
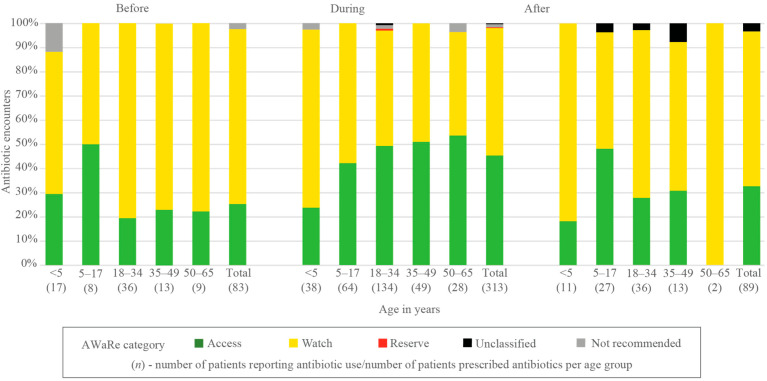
Distribution of antibiotic use before, during, and after outpatient visit across age groups, according to the WHO AWaRe classification by percentage of encounters.

**Figure 2 antibiotics-11-00574-f002:**
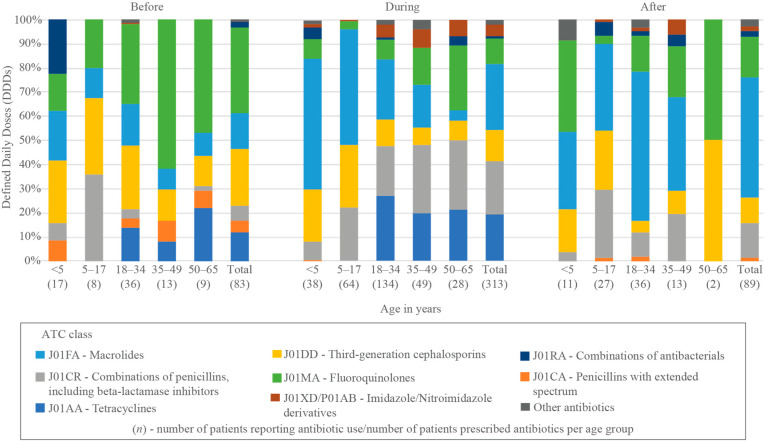
Distribution of antibiotic use before, during, and after outpatient visit, across age groups, according to WHO ATC classification by percentage of total Defined Daily Doses (DDDs).

**Figure 3 antibiotics-11-00574-f003:**
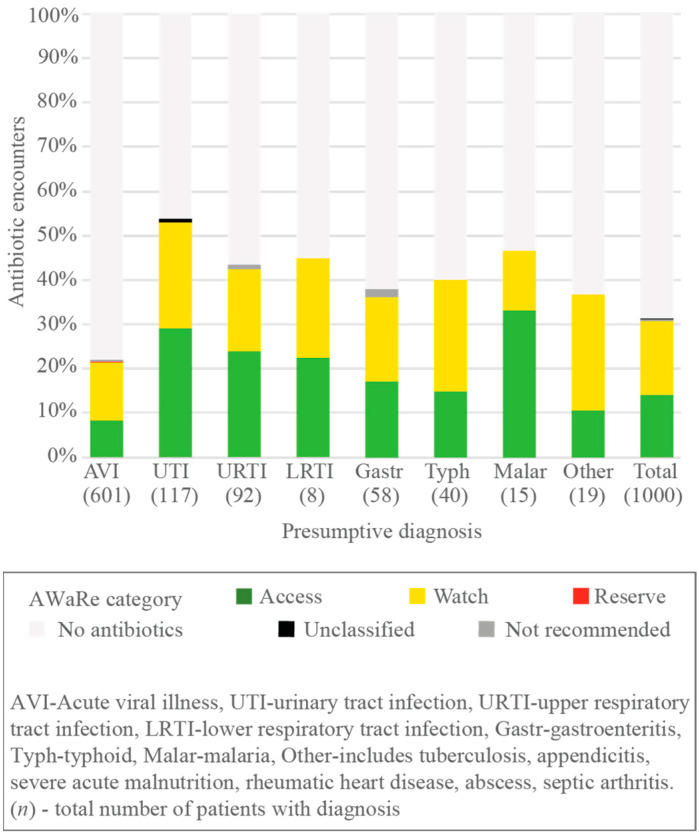
Distribution of antibiotic prescription during the outpatient visit by presumptive diagnosis for all patients and AWaRe category in encounters.

**Figure 4 antibiotics-11-00574-f004:**
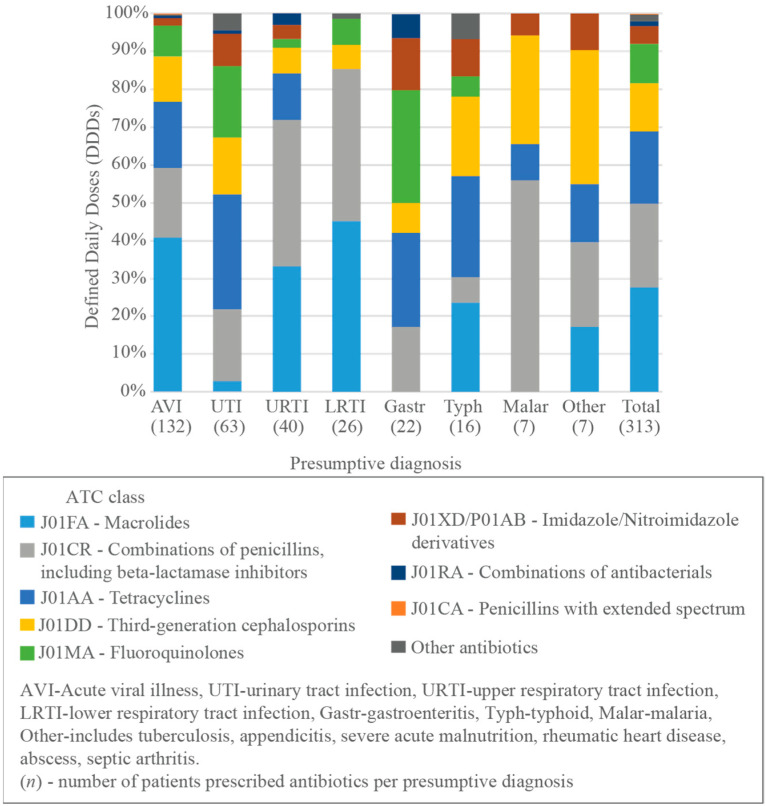
Distribution of antibiotic prescription during the outpatient visit by presumptive diagnosis, according to WHO ATC classification in percentage of total Defined Daily Doses (DDDs).

**Table 1 antibiotics-11-00574-t001:** Demographic and clinical characteristics of study participants by age group.

	Age Group in Years										Total	
	<5		5–17		18–34		35–49		50–65		<5–65	
Variable	(*n* = 183)		(*n* = 321)		(*n* = 298)		(*n* = 117)		(*n* = 81)		(*n* = 1000)	
Female sex, *n* (%)	82	(45)	120	(37)	146	(49)	60	(51)	40	(49)	448	(44.8)
Age in years, median (IQR)	2	(1–3)	10	(7–13)	23	(20–30)	40	(36–45)	56	(50–63)	17	(6–29)
Temperature ≥ 37.5 °C, *n* (%)	57	(31)	127	(40)	163	(55)	62	(53)	51	(63)	460	(46.0)
Fever duration in days, median (IQR)	3	(2–4)	3	(2–4)	4	(3–8)	4	(3–8)	5	(3–8)	3	(2–5)
Time to resolution of illness, median (IQR)	3	(2–3)	3	(2–3)	5	(4–6)	5	(5–8)	5	(4–5)	3	(3–5)
Presumptive diagnosis, *n* (%)												
Acute viral illness	123	(67)	197	(61)	177	(59)	60	(51)	44	(54)	601	(60.1)
Urinary tract infection	10	(5)	26	(8)	43	(14)	23	(20)	15	(19)	117	(11.7)
Upper respiratory tract infection	16	(9)	28	(9)	26	(9)	12	(10)	10	(12)	92	(9.2)
Lower respiratory tract infection	15	(8)	11	(3)	20	(7)	8	(7)	4	(5)	58	(5.8)
Typhoid	2	(1)	11	(3)	17	(6)	5	(4)	5	(6)	40	(4.0)
Malaria	0	(0)	2	(1)	7	(2)	5	(4)	1	(1)	15	(1.5)
Gastroenteritis	12	(7)	35	(11)	5	(2)	4	(3)	2	(2)	58	(5.8)
Other *	5	(3)	11	(3)	3	(1)	0	(0)	0	(0)	19	(1.9)

* Includes tuberculosis, appendicitis, severe acute malnutrition, rheumatic heart disease, abscess, and septic arthritis. (IQR)—interquartile range.

**Table 2 antibiotics-11-00574-t002:** Combined reported antibiotic use and antibiotic prescription throughout the course of illness.

Before	During	After	Total %, (*n*)	
8.3% (*n* = 83)	31.3% (*n* = 313)	8.9% (*n* = 89)	100% (*n* = 1000)	
+	-	-	4.3%	(43)
+	+	-	3.3%	(33)
+	-	+	0.5%	(5)
+	+	+	0.2%	(2)
-	+	+	3.3%	(33)
-	+	-	24.5%	(245)
-	-	+	4.9%	(49)
-	-	-	59.0%	(59)

+ reported antibiotic use or antibiotic prescription, - no antibiotics.

## Data Availability

The data presented in this study are available on request from the authors.

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
