# Peer review of "Antibiotic Use before, during, and after Seeking Care for Acute Febrile Illness at a Hospital Outpatient Department: A Cross-Sectional Study from Rural India"

_antibiotics, 2022, doi:10.3390/antibiotics11050574_

Round 1

Reviewer 1 Report

The paper concerns the "Antibiotic use before, during and after seeking care for acute febrile illness at a hospital outpatient department: a cross-setional study from rural India"

The paper contains valuable insight into combating the problem of antimicrobials resistance (AMR).

The reader will benefit if some minor amends highlighted below are effected.

The Introduction contains instructional text at lines 72-80

Add the symbol for Celsius degrees for temperature in Table 1

Also, in Table 1: for presumptive diagnosis, it would be informative to include aggregated/total proportion in addition to the age groups

Line 251, please clarify what you mean by DDD/total
Also, remove some instructional words from lines 372-379
Provide grants number in the grants section

The references listed have serial numbers twice. Please remove one and check that all references are uniform.

Additionally, can you please use DDD/per 1000 population per year to report the outcome of utilization? This is the standard international unit for comparability with other regions or countries.

Reviewer 2 Report

The authors Bronwen Holloway et al presented a very good study in this manuscript. However there are a lot of flaws in the manuscript, which needs proper rectification by the others. Some of these are mentioned as below:

1. In line 27 replace the word guardian by patient attendant.

2.Rephrase the line 27-29 in abstract section..

3. Line 64 needs rephrasing.

4.Kindly mention that how the authors confirmed that the patients were using the antibiotics and were showing adherence to its usage as per recommended scheduled. What was yours source of confirmation of compliance?

5. Add conclusion of the manuscript.

6. Check the manuscript for grammatical mistakes and revise accordingly.

Reviewer 3 Report

The topic is of interest, and the methodology/results included in this manuscript help understand the evaluation and monitoring of antibiotics considering the WHO AWaRe classification of antibiotics before, during and after seeking care. In addition, the discussion is very well written.

The manuscript is satisfactory for publishing. However, the reviewer includes some general comments: (It scares me the use of antibiotics for the “treatment” of the acute viral illness).

1) Please erase the lines – 72 to 80

2) Line 83: I propose to include the study period (June to August 2019)

3) Line 87: In summary, 533 antibiotic courses were reported across 41.0% of the 1000 patients. How has it been calculated? How long is a course of an antibiotic? (7 days ?); how many antibiotics are there in a course?

4) Discussion: norfloxacin is classified as “wastch.”

Reviewer 4 Report

Dear Authors, thank you for giving me the opportunity to review your manuscript. There are some comments that I would like to make with the intention of improving it: 

Major Comments:

  1. Introduction: Please, describe the current state of antimicrobial resistance rates in India, suggesting it could be due to the irrational antibiotic use.
  2. Although the Instructions for Authors suggest the “Materials and Methods” section to be put after “Results” and “Discussion” I guess it would be reasonable to put it after Introduction.
  3. Table 1: Please, add the “Total” column, showing % and n of females/presumptive diagnosis/IQRs for the whole study.
  4. Please, explain the abbreviations (IQR, DDD, FDC) when firstly mentioned in the text.
  5. Antibiotic use before seeking care: Does it mean that it is the group with a 100% self-medication of antibiotics patients? Does it mean that patients purchased the antibiotics over-the-counter? How did they choose the antibiotic? It should be explained in “Materials and methods” in details.
  6. Antibiotic prescription during the outpatient visits: There should be the mentioning that 33 of 83 patients who reported the use of antibiotics before the outpatient visit (39,8%) were prescribed antibiotics during the visit by the physician. Does it mean the 60% of irrational and excessive antimicrobial use? It should be also analyzed if the physician switch/change the initial antimicrobial (self-medication)?
  7. Line 131: Please specify the combinations of two antimicrobials prescribed with the presumptive diagnosis. Does it correspond to the Indian national treatment guidelines? This observation should be mentioned in the Discussion as well.
  8. Antibiotic treatment modification after the outpatient visits: Does it mean that it is the group with a 100% of non-compliance to the physician recommendations? What kind of modification that was (a switch from one antibiotic class to another or a modification of the drug within the antimicrobial class)? The percentage should be calculated for 33 of 313 who received the prescription of the physician (10,5%).

In other words, the observed patterns of behavior of patients (active consumers of antimicrobials, compliant patients that adhere to doctor’s recommendations, opponents of antimicrobial treatment, etc.) may be shown by the Table below and discussed appropriately.

AB use
before the visit

8,3% (n=83)

AB prescribed during the visit

31,3% (n=313)

Treatment modification after the visit

8,9% (n=89)

Total %, (n)

+

-

-

-

+

-

+

+

+

+

+

-

-

+

+

-

-

+

-

-

-

  1. Discussion Line 243: Specify the proportion of patients who did not receive any antibiotics for UTI and LRTI. If you suggest this could be due to the culture results, what is the average time of getting the microbiological results then? Is it more than the 7-days follow-up period of the Study?
  2. Materials and methods: The part of Data collection procedure before/during/after should be explained as mentioned above.
  3. References: Please check if the references are described according to the Instructions for Authors

Minor Comments:

(line 72-80) I guess this part is not needed

(line 82) change “reported” to “presented”

(line 104) change “fell” to “belonged to”

(line 214) change “the topic of antibiotic use” to “antimicrobial stewardship”

(line 218) change “is aligned with” to “corresponds to”

(line 228) delete extra spaces

(line 271) does the “surface” mean “soil”?

(line 313) population

(line 332) included

(line 362) fixed-dose combinations

Reviewer 5 Report

This is an interesting, patient—level descriptive DU study with a wealth of data from a low income setting. I appreciate the work behind data collection in the resource limited setting. Despite that I found the study aims very important and absolutely support the report of this study, I found several major issues that needs to be resolved before. I found it very hard to read the text due to the high number of numerical data in the text, which sometimes overlap with figures. In the present format the presentation of results is suboptimal. I provided a detailed comment to enable to improve it.

Major

How did you ensure that pre-visit antibiotic encounters and hospital outpatient clinic encounters were for the same episod. What time frame you used? Please clarify

The use of DDDs is problematic for pre and post visit antibiotic use derived from interviews. Secondly, as part large part of your population were children, where adult DDD cannot be applied, every analysis and comparison releted to AB use patterns is uninterpretable. You should use instead the number of encounters to compare pattern of use. Consequently, Fig 2 should be redone,

Please provide a flow chart (or a table?), where we can better follow what happened with patient groups in the pre, during and post visit period. It is very difficult to follow these numbers in the text. For example show in fow chart that out of the 83 previsit AB use, 33 received AB during the visit and 5 also received AB post visit. Similar for the no AB use pre-visit, etc…

The presumptive diagnosis groups has no sense in this form: URTI and gastroenteritis is acute viral illneses in majority of cases. What you included in the acute viral illness category than? Please list. I think this part should be re categorised…….. What you included in the LRTI category? Please also list. Also Malaria needs antimalarial treatment, so it is misconductiong to analyse its AB treatment…This should be deleted (mention in the limitation is not enough in my opinion)

How was time to resolution definied.  No info in the methods how it was calculated, etc.

line 107_108: do not count % for numbers less than 10,anddo not compare this…. also not ok to visualise this on Fig 1.

Fig 3. Malaria should be deleted, categorisation needs to be redone (see my previous comment). After, please provide some logical order for the columns (e.g access gradient)

line 244: meaningful LRTI culturing is very rare in AC even in western countries, so this explanation is not ok and should be considered for deletion.

line 245-246: what you mean access problems? Seems that access to AB is very easy in India…

Fig 4: In stead of DDD use the number of encounters, recategorise presumptive diagnoses. Finally provide ATC codes for the names to ensure clarity.

line 241: important new data, not mentioned in the results if I am right. Please add there (after recalculation for new categories)

line 252-254: depends on age/number/type of comorbidities. Dual treatment (beta lactam plus macrolide ) is ok in elderly population and/or certain comorbide population with CAP

MINOR

I missed results on how AB choices were changed during the illness. Have they changed access or watch categories to upper categories (from „pre” to „during” period, from „during” to „post” period). This is much more important info than comparing purely patterns

Unclear how many doctors prescription pattern is analysed in this study

Doxycyclin (plus metronidazol) for UTI? How do you explain this. Only good for urethritis. Is this included in your data?

How do you explain that a recent study from India found 12% of FDC use, while you only 2-3% Which one is more reliable and why?

Provide Total column for Fig 1.

Explan FDC at the first use…

sometimes you wrote < 5 , sometimes 2 months to 5 years, sometimes to 59 months. Use it consequently and explain how you handled e..e. 17-18 years.

line 202: illness or fever. I think fever.

line 221: refer to reference 19 has some bias as it only analysed per os products, this should be made clear. Also comparison to ref 20 is problematic (limited sense) as ref 20 only focused on severely ill patients.

line 221-223: This is a speculation, despite that this data can be very easily calculated from the dataset

line 273-275: what are these % (18, 16). Better to include this in the results.

line 322 vs line 330/331: controversy who prescribed

line 329 vs. 340: in line 329 you mention interview for all the 3 type of encounters…

Round 2

Reviewer 2 Report

Thanks for rectification of the concerns

Reviewer 5 Report

Author replies are ok for me.